# Combined PK/PD Index May Be a More Appropriate PK/PD Index for Cefoperazone/Sulbactam against *Acinetobacter baumannii* in Patients with Hospital-Acquired Pneumonia

**DOI:** 10.3390/antibiotics11050703

**Published:** 2022-05-23

**Authors:** Yingjie Zhou, Jing Zhang, Yuancheng Chen, Jufang Wu, Beining Guo, Xiaojie Wu, Yingyuan Zhang, Minggui Wang, Ru Ya, Hao Huang

**Affiliations:** 1Institute of Antibiotics, Huashan Hospital, Fudan University, Shanghai 200040, China; 13916425190@163.com (Y.Z.); zhangj_fudan@aliyun.com (J.Z.); 13816357099@163.com (J.W.); 13045666468@163.com (B.G.); maomao_xj@163.com (X.W.); yyzhang@hsh.stn.sh.cn (Y.Z.); mgwang@fudan.edu.cn (M.W.); 2Key Laboratory of Clinical Pharmacology of Antibiotics, National Health Commission, Shanghai 200040, China; 3National Clinical Research Center for Aging and Medicine, Huashan Hospital, Fudan University, Shanghai 200040, China; 4Phase I Unit, Huashan Hospital, Fudan University, Shanghai 200040, China; 5Yonghe Branch of Huashan Hospital, Fudan University, Shanghai 200436, China; yaru3000@126.com (R.Y.); huanghao97224007@126.com (H.H.)

**Keywords:** cefoperazone, sulbactam, pharmacokinetic/pharmacodynamic, combined PK/PD index, hospital-acquired pneumonia, *Acinetobacter baumannii*

## Abstract

Cefoperazone/sulbactam (CPZ/SUL) is a β-lactam and β-lactamase inhibitor combination therapy for the treatment of respiratory tract infections. Using data from a prospective, multiple-center, open-label clinical trial in 54 patients with hospital-acquired pneumonia or ventilator-associated pneumonia caused by multidrug-resistant *Acinetobacter baumannii* (Ab), we showed that a combined PK/PD index %(T > MIC_cpz_*T > MIC_sul_) is a more appropriate PK/PD index against Ab, compared to the PK/PD index (%T > MIC) for a single drug. For a 2 h infusion, the PK/PD cutoff of CPZ/SUL (2 g/1 g, q8h) for clinical and microbiological efficacy was 4/2 and 1/0.5 mg/L, respectively. The corresponding cumulative fraction of response was 46.5% and 25.3%, respectively. Results based on the combined PK/PD index were quite similar to that based on the joint probability of target attainment. The two drugs have interaction from the viewpoint of PK/PD. When the dose of one drug was too high, the PK/PD cutoff was often determined by another drug in which the dose was maintained. In most cases, sulbactam exerted the main effect against infection by Ab in the complex CPZ/SUL, which was similar to the literature reports. When the MIC of CPZ was 8, 16, or 32 mg/L, a CPZ/SUL 2 g/1 g (q8h), 2 g/2 g (q8h), or 2 g/2 g (q6h) (infusion was all 3 h) was recommended, respectively. A clinical efficacy and safety study to confirm simulation results is warranted.

## 1. Introduction

Multi-drug resistant (MDR) and pan-drug resistant (PDR) bacteria are becoming more common due to the wide use of antibiotics and most of these bacteria include non-fermentative bacilli [1,2,3]. A multi-center epidemiological survey showed that non-fermentative bacilli are the most common pathogen for hospital-acquired pneumonia (HAP) in China, and *Acinetobacter baumanii* (Ab) and *Pseudomonas aeruginosa* (PA) constitute up to 30% and 22%, respectively [4]. Due to the lack of effective antibiotics for the treatment of infections by MDR or PDR non-fermentative bacilli, HAP has become a significant public health threat.

Cefoperazone/sulbactam (CPZ/SUL) is a β-lactam and β-lactamase inhibitor combination which has been selected as alternative drugs for the treatment of infection by non-fermentative bacilli. Cefoperazone exerts a bactericidal effect by inhibiting the synthesis of the cell wall of bacteria [5]. Surveillance studies showed that PA or Ab were predominantly sensitive to CPZ/SUL [6]. Hence, this drug is recommended as one of the main anti-infective agents for the treatment of HAP [7].

Pharmacokinetics (PK) in patients may be different in comparison to that in healthy volunteers because some covariates such as disease occurrence and concomitant medications may affect PK behavior. Population pharmacokinetics (PPK) is the study of variability in drug concentrations within a patient population receiving clinically relevant doses of a drug of interest [8]. This is important for successfully interpreting the clinical and microbiological efficacy of the drug. Although there are some PK reports of CPZ/SUL in patients with renal dysfunction [9], continuous ambulatory peritoneal dialysis [10], acute appendicitis [11], and in seriously ill elderly patients [12], there are no studies to date investigating the PK of CPZ/SUL in HAP patients. Meanwhile, there are two PPK reports of CPZ in children [13,14], and there is no PPK report of CPZ/SUL in adults.

Several reports showed that there is synergy between CPZ and SUL against Ab [15] or PA [16]. However, only the pharmacokinetic/pharmacodynamic (PK/PD) index for SUL was reported [17]. Whether it is suitable for CPZ/SUL combination is still unknown.

In recent years, the joint probability of target attainment (PTA) was introduced for the PK/PD analysis of β-lactam/β-lactamase inhibitors complex [18,19,20]. Joint PTA means PTA based on the simultaneous achievement of the individual PK/PD targets [21]. To calculate joint PTA, the PK/PD target for each drug should be obtained first. So, this method still does not consider the compound as a whole (i.e., does not calculate the PK/PD target for the compound). Whether joint PTA is suitable for CPZ/SUL combination remains unknown.

The objectives of this study were to (1) describe PPK and pharmacodynamic (PD) of CPZ/SUL in HAP patients; (2) find the best PK/PD index predicting clinical and microbiological efficacy of CPZ/SUL, and (3) optimize dosing regimen using the Monte Carlo simulation.

## 2. Results

### 2.1. Baseline Characteristics of Patients

Thirty-nine male (72.2%) and fifteen female patients (27.8%) were enrolled (Table 1). The mean age was 46 years. The main concomitant diseases were traumatic brain injury (44.4%) and hypertension (24.1%). Most frequent concomitant medications were ambroxol (83%) and sodium valproate (50%), and main concomitant antifungal was fluconazole (15%). Before administration of CPZ/SUL, most patients were given aminoglycosides (34.5%, such as amikacin), cephalosporins (34.5%, such as ceftriaxone), and quinolones (20%, such as levofloxacin). Twenty-nine patients (53.7%) and twenty-two patients (40.7%) were infected by *Acinetobacter spp.* and PA, and the other 3 patients were infected by both. In vitro activity of CPZ/SUL is summarized in Appendix A.

### 2.2. PK, PD, and Safety of Cefoperazone and Sulbactam

Time profiles of CPZ and SUL are shown in Figure 1. The C_max_ of cefoperazone in the q8h group (infusion time = 1.5 h) was 120 mg/L. Corresponding T_1/2_ and V_d_ were 4.49 h and 20.9 L, respectively. The C_max_ of sulbactam in the q8h group (infusion time = 1.5 h) was 27.4 mg/L, and T_1/2_ was 1.82 h.

The clinical and microbiological response rates of CPZ/SUL in patients with Ab infection were 78.1% (25/32) and 71.9% (23/32), respectively. For patients with PA infections, the clinical response rate was 70.8% (17/24). Two patients (3.6%) experienced a skin rash when receiving CPZ/SUL on the 5th and 6th day, which may be drug related. They were recovered after stopping therapy and treated with antihistamines. No laboratory adverse event (AE) or serious AE was observed.

### 2.3. Population Pharmacokinetics

PK of CPZ was consistent with the two-compartment model. Inter-individual variability (IIV) of clearance (CL) and the distribution volume in the central (V_1_) and peripheral compartment (V_2_) were described using an exponential error model. Final estimates of parameters were shown in Table 2. High blood pressure had significant impact on V_2_ (ΔOFV = −7.25, *p* = 0.007). Two separate proportional residual errors were introduced because they decreased OFV significantly (ΔOFV = −25.1, *p* < 0.001).

PK of SUL was also consistent with the two-compartment model (Table 2). Age, in combination with baclofen, was a significant covariate. Age showed significant effects on CL (ΔOFV = −17.1, *p* < 0.001), while baclofen had significant effects on Q (ΔOFV = −10.5, *p* = 0.001).

Concentration data were well described by the PPK model (Appendix A). The linear regression line for observations (DV) vs. individual predictions (IPRE) was consistent with the identity line. The distribution of data points was uniform across zero horizontal lines in the plot of conditional weighted residue (CWRE) vs. population predictions (PRED) or time (Appendix A).

Most of the observed concentration data fell within the 90% confidence interval (CI) of simulated values from the PPK model (Appendix A). There were 15.0% and 8.2% data out of limits for CPZ and SUL, respectively. During bootstrap validation, 296 (98.7%) and 298 calculations (98.3%) had successful minimization for CPZ and SUL, respectively. The mean relative deviation of parameter estimates between the bootstrap and original datasets was both 2.1% for CPZ and SUL (Table 2).

Compared to patients with no high blood pressure, T_1/2_, C_min,_ and V_d_ of cefoperazone in patients with high blood pressure increased by 17% (3.5 vs. 3.0 h), 15% (28 vs. 24 mg/L), and 11% (17 vs. 15 L), respectively. The impacts of high blood pressure on other PK parameters were weak (relative deviation < 10%). Age had an impact on the PK parameters of sulbactam. When age increased, AUC, C_max_, T_1/2,_ and C also increased, while CL and V_d_ reduced. Compared to patients aged 46, C_min_ and AUC_0-inf_ decreased by 63% (0.36 vs. 0.97 mg/L) and 32% (48 vs. 69 h·mg/L) in patients with aged 18, respectively. Meanwhile, they increased by 212% (3.04 vs. 0.97 mg/L) and 66% (116 vs. 69 h·mg/L) in patients aged 70, respectively. Compared to patients with no concomitant medication of baclofen, the terminal elimination rate of sulbactam increased by 6.6% (0.62 vs. 0.58 1/h), while C_max_ and V_d_ reduced by 4.5% (28.0 vs. 29.4 mg/L) and 3.8% (26.6 vs. 27.7 L) in patients with concomitant medication of baclofen, respectively. The impact of baclofen on other PK parameters was quite weak (relative deviation < 2%).

### 2.4. PK/PD Analysis

#### 2.4.1. Effect of Covariate on PK/PD of CPZ/SUL

The effect of high blood pressure on %T > MIC of cefoperazone against Ab was weak (Appendix A). Compared to patients with no high blood pressure, %T > MIC increased only by 1% in patients with high blood pressure (85.5% vs. 84.5%). The impact of taking baclofen on %T > MIC of sulbactam was also weak (Appendix A). When age was 46 years, %T > MIC was 42.9% or 41.8% in patients with or without administration of baclofen, respectively. In contrast, age had a significant impact on %T > MIC of sulbactam. For patients who did not take baclofen, %T > MIC was 31%, 42%, and 58% in patients aged 18, 46, and 70 years, respectively.

#### 2.4.2. Analysis Based on PK/PD Index for a Single Drug

The correlation between the PK/PD index of cefoperazone and clinical efficacy was shown in Figure 2. All PK/PD indices have a positive correlation with clinical efficacy. The *p*-value of the slope obtained from the logistic regression for Ln(AUC_0–24_/MIC), Ln(C_max_/MIC), and %T > MIC were 0.049, 0.060, and 0.0595, respectively. The PK/PD target of cefoperazone against Ab was shown in Table 3. For clinical efficacy, the target of %T > MIC and AUC_0–24_/MIC was 54.8% and 44.3, respectively. The mean probability of successful clinical efficacy was 89% when the PK/PD index ≥ target. For microbiological efficacy, %T > MIC and AUC_0–24_/MIC targets were 83.2% and 162.4, respectively. The corresponding probability of successful microbiological efficacy when the PK/PD index ≥ target was 79% and 91%, respectively.

The correlation between the PK/PD index of sulbactam and clinical efficacy was similar to that of cefoperazone (Figure 2). The *p*-value of the slope obtained from the logistic regression for Ln(AUC_0–24_/MIC), Ln(C_max_/MIC), and %T > MIC were 0.038, 0.057, and 0.044, respectively. For infection by Ab, the clinical target of %T > MIC and AUC_0–24_/MIC were 36.6% and 23.3, respectively (Table 3). The probability of successful clinical efficacy was 94% when the PK/PD index ≥ target. The microbiological target of %T > MIC and AUC_0–24_/MIC were 61.1% and 50.4, respectively. The probability of successful microbiological efficacy was 90% when PK/PD index ≥ target.

The PK/PD cutoff based on the PK/PD index for a single drug was summarized in Table 4, Appendix A. For cefoperazone (regimen 2 g q8h), the PK/PD cutoff of %T > MIC was 16 mg/L when the infusion time was ≤ 2 h (Table 4). It increased to 32 mg/L when infusion time was 3–4 h. For sulbactam (regimen 1 g q8h), the PK/PD cutoff of %T > MIC was 1–2 mg/L when the infusion time was ≤ 2 h. It elevated to 4 mg when the infusion time was 3–4 h. When the PK/PD index was AUC_0–24_/MIC, the PK/PD cutoff for cefoperazone (2 g q8h) and sulbactam (1 g q8h) was 16 and 4 mg/L, respectively.

When the infusion time was 0.5–2 h, based on %T > MIC, the PK/PD cutoff of cefoperazone was 8 and 16 mg/L for doses of 1 g and 1.5 g, respectively (Appendix A). PK/PD cutoff increased to 32 mg/L when the dose increased to 3–4 g. The PK/PD cutoff was 64 mg/L when the cefoperazone dose was 6 g. Based on AUC_0–24_/MIC, the PK/PD cutoff of cefoperazone was 8, 16, and 32 mg/L for doses of 1–1.5 g, 3 g, and 4–6 g, respectively.

When the infusion time was 0.5–2 h, based on %T > MIC, the PK/PD cutoff of sulbactam was 2–4 mg/L for a dose of 1.5 g (Appendix A). When the sulbactam dose increased to 3 g, the PK/PD cutoff increased to 4–8 mg/L. Based on AUC_0–24_/MIC, the PK/PD cutoff of sulbactam was 4 and 8 mg/L for doses of 1.5 g and 2–3 g, respectively.

The cumulative fraction of response (CFR) based on PK/PD index for cefoperazone was summarized in Appendix A. Based on %T > MIC, when the infusion time was 0.5–2 h: CFR was 74–78% for a dose of 2 g; when the dose increased to 3 g, CFR was 84–87%; when the dose was 6 g, CFR was 94–95%. Based on AUC_0–24_/MIC, the CFR of cefoperazone was 67%, 78%, and 92% for doses of 2 g, 3 g, and 6 g, respectively.

The CFR based on the PK/PD index for sulbactam was shown in Appendix A. Based on %T > MIC, when the infusion time was 0.5–2 h: CFR was 38–49% and 48–60% for doses 1 g and 1.5 g; CFR was 55–68% and 66–79% for a dose of 2 g and 3 g, respectively. Based on AUC_0–24_/MIC, the CFR of sulbactam was 49%, 61%, 69%, and 79% for doses of 1 g, 1.5 g, 2 g, and 3 g, respectively.

#### 2.4.3. Analysis Based on Combined PK/PD Index

The construction of the combined PK/PD index and its correlation vs. clinical efficacy against Ab is shown in Appendix A. The various combinations between %T > MIC, AUC_0–24_/MIC, and Ln(AUC_0–24_/MIC) for two drugs were tried. For %(T > MIC_cpz_*T > MIC_sul_), the P_logstic_ (*p*-value for logistic regression) and P_cross_ (*p*-value for cross tabulation) were 0.047 and 0.020, respectively. For AUC_0–24_/MIC_cpz_*AUC_0–24_/MIC_sul_, P_logstic_ and P_cross_ were 0.299 and 0.075, respectively. The correlation between Ln(AUC_0–24_/MIC)_cpz_*Ln(AUC_0–24_/MIC)_sul_ and clinical efficacy was better than that of AUC_0–24_/MIC_cpz_*AUC_0–24_/MIC_sul_: P_logstic_ and P_cross_ were 0.057 and 0.030, respectively.

Correlation between the combined PK/PD index of CPZ/SUL and clinical or microbiological efficacy is shown in Figure 3. %(T > MIC_cpz_*T > MIC_sul_) had a positive correlation with clinical or microbiological efficacy. P_logstic_ was 0.047 and 0.114, respectively. The clinical and microbiological target of %(T > MIC_cpz_*T > MIC_sul_) was 36.6% and 61.1% (Table 5). Probability for successful efficacy was 94% and 90% when %(T > MIC_cpz_*T > MIC_sul_) ≥ target, respectively. Ln(AUC_0–24_/MIC)_cpz_*Ln(AUC_0–24_/MIC)_sul_ also had a positive correlation with efficacy. P_logstic_ was 0.057 and 0.091 for clinical and microbiological efficacy, respectively. The clinical and microbiological target of Ln(AUC_0–24_/MIC)_cpz_*Ln(AUC_0–24_/MIC)_sul_ was 14.37 and 19.75. The probability for successful efficacy was 94% and 91% when Ln(AUC_0–24_/MIC)_cpz_*Ln(AUC_0–24_/MIC)_sul_ ≥ target, respectively.

The PK/PD cutoff based on %(T > MIC_cpz_*T > MIC_sul_) against Ab is summarized in Table 6. For clinical efficacy, the cutoff was 2–4 mg/L for CPZ/SUlL (1 g/1 g) when the infusion time was 0.5–2 h. The PK/PD cutoff for CPZ/SUL (2~6 g/1 g) was similar to that for CPZ/SUlL (1 g/1 g). For CPZ/SUL (1.5 g/1.5 g and 2 g/2 g), the PK/PD cutoff increased to 4–8 mg/L when infusion time was 0.5–2 h. The PK/PD cutoff was 8–16 mg/L for CPZ/SUL (2 g/3 g) when infusion time was 0.5–2 h. As for microbiological efficacy, the change of PK/PD cutoff along with dose was similar to that for clinical efficacy. The PK/PD cutoff for CPZ/SUL (2~6 g/1 g) was 0.5–1 mg/L when the infusion time was 0.5–2 h. Cutoff increased to 1–2 and 2–4 mg/L for CPZ/SUL (2 g/2 g) and (2 g/3 g), respectively.

The CFR of %(T > MIC_cpz_*T > MIC_sul_) against Ab is shown in Appendix A. For clinical efficacy, CFR was 36–46% for CPZ/SUL (2 g/1 g) when infusion time was 0.5–2 h. CFR increased to 49–61% when the CPZ/SUL dose was 2 g/2 g. When the CPZ/SUL dose was 2 g/3 g, CFR was 56–67% when infusion time was 0.5–2 h. The impact of the cefoperazone dose on CFR was weak compared to that of the sulbactam dose: when infusion time was 0.5 h, CFR was 32% and 38% for CPZ/SUL 1 g/1 g and 6 g/1 g, respectively. When infusion time was 2 h, the corresponding CFR was 42% and 49%, respectively. For microbiological efficacy, the impact of the CPZ and SUL dose on CFR was similar to that of clinical efficacy: when infusion time was 2 h, CFR increased from 25% to 38% when the dose of CPZ/SUL increased from 2 g/1 g to 2 g/2 g; CFR only increased from 25% to 26% when the dose of CPZ/SUL increased from 2 g/1 g to 6 g/1 g.

#### 2.4.4. PK/PD Analysis Based on Joint PTA

As both %T > MIC and AUC_0–24_/MIC had good correlation with clinical or microbiological efficacy (Figure 2), two equations for the calculation of joint PTA were constructed: PTA(%T > MIC_cpz_)*PTA(%T > MIC_sul_), PTA(AUC_0–24_/MIC_cpz_)*PTA(AUC_0–24_/MIC_sul_). The cutoff based on PTA(%T > MIC_cpz_)*PTA(%T > MIC_sul_) is shown in Table 7. Based on clinical efficacy, the cutoff was 2–4 mg/L for CPZ/SUL (2 g/1 g) when the infusion time was 0.5–2 h. When the dose of CPZ/SUL increased to 2 g/2 g, the cutoff increased to 4–8 mg/L. The PK/PD cutoff for CPZ/SUL (2 g/3 g) was 8–16 mg/L when the infusion time was 0.5–2 h. The cutoff for CPZ/SUL (4~6 g/1 g) was the same as that of CPZ/SUL (2 g/1 g). For microbiological efficacy, the PK/PD cutoff was 0.5–1 mg/L for CPZ/SUL (2~6 g/1 g) when the infusion time was 0.5–2 h. The corresponding cutoff for CPZ/SUL 2 g/2 g and 2 g/3 g were 1–2 and 2 mg/L, respectively.

The cutoff based on PTA(AUC_0–24_/MIC_cpz_)*PTA(AUC_0–24_/MIC_sul_) is shown in Appendix A. For CPZ/SUL(2 g/1 g), the cutoff was 8 and 4 mg/L for clinical and microbiological efficacy, respectively. For CPZ/SUL 2 g/2 g and 2 g/3 g, the cutoff was both 16 and 4 mg/L for clinical and microbiological efficacy. The cutoff for CPZ/SUL (4~6 g/1 g) was the same as that of CPZ/SUL (2 g/1 g).

## 3. Discussion

In this study, the majority of the patients either had a brain injury (44%) or a cerebral hemorrhage (20%) (Table 1). The reason was as follows: (1) Nosocomial infection is a common complication of cerebral hemorrhage. Thirty-six percent of patients in neural ICU (stay time > 48 h) have the complication of infection, where pneumoniae is the most common [22]. Prof. Archana Hinduja et al. performed a study to investigate the incidence rate of nosocomial infection in patients with intracerebral hemorrhage, as well as risk factors and prognosis. Results showed that 26% of patients have at least one infection, where the most common was pneumoniae (18%) [23]; (2) severe craniocerebral trauma has been regarded as the risk factor for hospital-acquired and ventilator-acquired pneumoniae [24].

The PK of CPZ and SUL were consistent with the two-compartment model, being consistent with previous reports [11]. Compared to PK results in healthy volunteers [25], T_1/2_ of CPZ and SUL in HAP patients increased by 94% (3.1 vs. 1.6 h) and 45% (1.4 vs. 1.0 h), while CL of CPZ decreased by 17% (78.8 vs. 95 mL/min). AUC_0-inf_ of CPZ and SUL was elevated by 29.2% (459.8 vs. 356 mg·h/L) and 6.5% (68.2 vs. 64 mg·h/L), respectively. V_ss_ of CPZ and SUL was elevated by 110% (21.4 vs. 10.2 L) and 77% (31.9 vs. 18.0 L), respectively. This indicated that both exposure and distribution of CPZ/SUL in HAP patients were enhanced, which may be beneficial for drug therapy.

The minimization algorithm had impacts on the base model and covariate screening during PPK analysis. For cefoperazone, when using the first-order conditional estimation (FOCE) method, concomitant medication with topiramate was a covariate on inter-compartment clearance. Red blood cell count and age were also covariates. After using the first-order conditional estimation with interaction (FOCEI) method, these factors did not appear as covariates after screening. Instead, high blood pressure was found to have a significant impact on V_2_. For sulbactam, when using the FOCE method, the intraindividual error was consistent with the additive model. Body mass index (BMI) and fluconazole were covariates on CL. After using the FOCEI method, the model for intra-individual error changed into a mixed type. Meanwhile, BMI and fluconazole were not significant covariates after the screening. Our explanation is as follows: FOCE does not consider the interaction between intra- and inter-individual variability. However, FOCEI considers this interaction and performs calculations [26,27]. The FOCEI method could improve the model fit because it could explain part of the variability in the data. Hence, for the same covariate, compared to the FOCE method, the reduction of OFV was different when using the FOCEI method. This could result in different significant covariates in the PPK study.

Results showed that high blood pressure has significant impact on V_2_ of cefoperazone. The explanation is as follows: (1) Cefoperazone is a lipophilic drug. It could distribute to the peripheral compartment (i.e., peripheral tissue) easily; (2) cerebral hemorrhage is one of the main complications of high blood pressure [28]. One study also showed that high blood pressure is a strong risk factor for hemorrhagic stroke [29]. In this study, many patients had cerebral hemorrhage (20%) or brain injury (44%) (Table 1). So, these diseases may destroy the barrier between blood and peripheral tissue. Hence, cefoperazone could easily reach peripheral tissue in patients with high blood pressure.

Sulbactam is a water-soluble drug. After administration, approximately 75–84% of the dose was excreted by the kidney [30,31]. When age increases, the renal function decreases. Therefore, the clearance of sulbactam also decreases. This could explain why age has a significant impact on the clearance of sulbactam. This was consistent with literature reports: a study in Thailand showed that age and CL_cr_ are covariates on CL of sulbactam, while hemoglobin was a covariate on V_1_ [32]. Another work showed that age had a significant impact on PK of sulbactam [33]: compared to younger (20–40 years) or middle-aged subjects (41–64 years), the AUC (*p* < 0.05), C_max,_ and T_1/2_ (*p* < 0.05) of sulbactam was the greatest in elderly subjects (65–85 years), while the CL was the lowest (*p* < 0.05).

A study showed that baclofen and sulbactam are substrates of an efflux transporter (multi-drug resistant protein 4, MRP4) [34]. MRP4 expression is high in the kidney [35]. Both baclofen and SUL are primarily eliminated by the kidney [31,36], hence baclofen may increase SUL concentration by combining with MRP4 competitively.

Because PA could easily colonize in the respiratory tract of patients [37], the rate of success of microbiological efficacy was quite low (20.8%, 5/24). Therefore, we did not perform exposure-response analysis (e.g., %(T > MIC_cpz_*T > MIC_sul_) vs. microbiological efficacy) and Monte Carlo simulation (MCS) in patients with PA infection.

The relationship between %*f*T > MIC of CPZ/SUL and efficacy was investigated. Percentage of correct classification using %*f*T > MIC was lower than that using %T > MIC (78.1% vs. 92.9%). PTA and CFR using %*f*T > MIC target were significantly lower than that using %T > MIC target. Hence, %T > MIC was employed as a variable during logistic regression.

PK/PD analysis showed that AUC_0–24_/MIC has the best correlation with clinical efficacy, followed by %T > MIC and C_max_/MIC (Figure 2). For CPZ, the *p*-value was 0.049 (AUC_0–24_/MIC) and 0.0595 (%T > MIC). For SUL, the *p*-value was 0.038 (AUC_0–24_/MIC) and 0.044 (%T > MIC). This may be relative to the short half-life of CPZ/SUL, especially SUL (T_1/2_ of CPZ/SUL was 3.1 h/1.4 h). As shown in Appendix A, both the P_logistic_ and P_cross_ for %(T > MIC_cpz_*T > MIC_sul_) was minimal. The P_logistic2_ for %(T > MIC_cpz_*T > MIC_sul_) was also minimal. Therefore, %T > MIC is the best PK/PD index for a single drug compared to AUC_0–24_/MIC.

As shown in Table 4, the PK/PD cutoff based on a single drug cefoperazone was obviously higher than that based on sulbactam. If using %T > MIC as the PK/PD index, the PK/PD cutoff for cefoperazone was 16 mg/L when the infusion time was 2 h. In contrast, the PK/PD cutoff for sulbactam was 2 mg/L. If using AUC_0–24_/MIC as a PK/PD index, the difference between the PK/PD cutoff between cefoperazone and sulbactam was also obvious. Hence, the PK/PD index for a single drug is not the best index for characterizing the PK/PD of CPZ/SUL.

As shown in Table 6, based on CPZ/SUL, the effect of adding CPZ is mild. For clinical efficacy, compared to CPZ/SUL 1 g/1 g, adding a CPZ dose of 2 g–6 g only increased the PK/PD cutoff from 2 to 4 mg/L when infusion time was 1 h. When infusion time was 0.5, 2, 3, or 4 h, adding a CPZ dose did not change the PK/PD cutoff. In contrast, the effect of adding SUL was obvious. For clinical efficacy, compared to CPZ/SUL 2 g/1 g, adding a SUL dose to 2 g increased the PK/PD cutoff from 4 to 8 mg/L when the infusion time was 2 h. Adding a SUL dose of 3 g further increased the PK/PD cutoff to 16 mg/L. For microbiological efficacy, when the infusion time was 2 h, adding an SUL dose of 1 g or 2 g could increase the PK/PD cutoff to 2 or 4 mg/L, respectively. This indicated that the main material basis is sulbactam. This is also shown in Table 7. Table 5 also indicated this: the PK/PD target based on %(T > MIC_cpz_*T > MIC_sul_) was same as that for %T > MIC target of sulbactam.

As shown in Table 6 and Table 7, the PK/PD cutoff based on the combined PK/PD index %(T > MIC_cpz_*T > MIC_sul_) was almost the same as that based on the combined PTA. For clinical efficacy, when the regimen of CPZ/SUL was 2 g/1 g (q8h), the PK/PD cutoff was 2, 4, and 4 mg/L when infusion time was 0.5, 1, 2 h, respectively (Table 6, Line 2). This was same as that based on PTA(%T > MIC_cpz_)*PTA(%T > MIC_sul_) (Table 7, Line 1). When the dose of CPZ/SUL was 2 g/2 g, for clinical efficacy, the PK/PD cutoff was 4–8 mg/L when the infusion time was 0.5–2 h (Table 6, Line 4). This was the same as that based on PTA(%T > MIC_cpz_)*PTA(%T > MIC_sul_) (Table 7, Line 2). When the dose of CPZ/SUL was 2 g/3 g, the PK/PD cutoff based on %(T > MIC_cpz_*T > MIC_sul_) (Table 6, Line 5) was almost the same as that based on PTA(%T > MIC_cpz_)*PTA(%T > MIC_sul_) (Table 7, Line 3). This indicated that the combined PK/PD index could be used for the PK/PD characterization of CPZ/SUL.

The CPZ and SUL had interaction from the viewpoint of the PK/PD. As shown in Table 8, for q8h dosing, when the PK/PD index was %T > MIC, the PK/PD cutoff for the CPZ/SUL regimen 2 g/1 g, 2 g/2 g was identical to that based on single drug sulbactam. When the dosing regimen was 2 g/3 g, the PK/PD cutoff was close to that based on sulbactam. When the PK/PD index was AUC_0–24_/MIC, for clinical efficacy, the PK/PD cutoff based on the combined PTA was identical to that based on sulbactam when the CPZ/SUL regimen was 2 g/1 g. When the regimen became 2 g/2 g or 2 g/3 g, the PK/PD cutoff based on combined PTA was identical to both SUL or CPZ. In contrast, for microbiological efficacy, the PK/PD cutoff based on combined PTA was identical to CPZ instead of SUL when the CPZ/SUL regimen was 2 g/2 g or 2 g/3 g. This indicated that when the dose of SUL was too high (e.g., 3 g), the PK/PD cutoff based on the combined PTA had a tendency to be close to that based on CPZ. When the dose of CPZ was too high (e.g., 4 g-6 g), the PK/PD cutoff based on combined PTA was identical to that based on SUL. These results were also similar to that when the dosing frequency was q6h or q12 h (Appendix A). Hence, when the dose of one drug was too high, the PK/PD cutoff was often determined by another drug in which the dose was maintained.

The fitting effect using %T > MIC of CPZ (%T > MIC_cpz_), %T > MIC of SUL (%T > MIC_sul_), or %(T > MIC_cpz_ × T > MIC_sul_) as the PK/PD index were compared. For patients with Ab infection, the Akaike information criterion (AIC) for the logistic regression model (clinical efficacy vs. %T > MIC_cpz_, %T > MIC_sul_ or %(T > MIC_cpz_ × T > MIC_sul_)) were 34.37, 33.37 and 32.93, respectively. For infection by PA, AIC for the above models were 23.4, 18.2 and 18.4, respectively. Only %(T > MIC_cpz_ × T > MIC_sul_) remained in the final model when %T > MIC_cpz_, %T > MIC_sul_, and %(T > MIC_cpz_ × T > MIC_sul_) were put together during multivariate logistic regression. Then, the fitting effect using the additive model (%T > MIC_cpz_ + %T > MIC_sul_), multiplicative model %(T > MIC_cpz_ × T > MIC_sul_), or exponential model ((T > MIC_cpz,ind_/T > MIC_cpz,mean_)^γ_cpz^ × (T > MIC_sul,ind_/T > MIC_sul,mean_)^γ_sul^) were compared. Multiplicative model had lower AIC compared to that using additive model. Although the exponential model had a bit lower sum of square of residuals compared to multiplicative model, the former was too complex for application in MCS. Therefore, we considered that product of %T > MIC of CPZ and SUL is an appropriate PK/PD index predicting efficacy of the CPZ/SUL combination. This reflects synergy between CPZ and SUL [31].

During the PK/PD analysis, using %T > *m**MIC as the PK/PD index (*m* = 1,2,3,…, *m* was selected when AIC was the lowest) was also tried because some literature reported that the best PK/PD indexes for β-lactams are %T > 4MIC in critically ill patients [38,39]. For infection by Ab, %T > 2MIC of CPZ or %T > MIC of SUL had the lowest *p*-value of the slope (0.047 and 0.066) in predicting clinical efficacy. Then %(T > 2MIC_cpz_ × T > MIC_sul_) was tried as the PK/PD index. The PK/PD cutoff based on %(T > 2MIC_cpz_ × T > MIC_sul_) was lower than that for %(T > MIC_cpz_ × T > MIC_sul_). PTA and CFR based on %(T > 2MIC_cpz_ × T > MIC_sul_) were also lower than that for %(T > MIC_cpz_ × T > MIC_sul_). Other similar results were obtained for the combination index %(T > *m**MIC_cpz_ × T > *m*’*MIC_sul_). Hence, the product of %T > MIC of CPZ and SUL was selected as the optimal PK/PD index.

According to MCS results, for treatment of infection by Ab, q8h dosing of CPZ/SUL (2 g/1 g) (infusion 3 h) was recommended for infection by Ab with MIC ≤ 8 mg/L. When MIC was 16 mg/L, a q8h regimen of CPZ/SUL (2 g/2 g) (infusion 3 h) was recommended. When MIC elevated to 32 mg/L, a q6h regimen of CPZ/SUL (2 g/2 g) (infusion 3 h) was recommended. When MIC was 64 mg/L, a q6h regimen of CPZ/SUL (2 g/3 g) (infusion 4 h) was recommended since CFR of %(T > MIC_cpz_ × T > MIC_sul_) was the highest (85.8%).

For the q8h regimen, when the infusion time was 3 h, adding 1 g of SUL into CPZ/SUL (2 g/1 g) not only increased the PK/PD cutoff from 8 to 16 mg/L (Table 6) but also increased the maximal CFR from 54% to 68% (Appendix A). A recent report showed that CPZ/SUL (2 g/2 g, q12h) could be used for HAP treatment, as 87.3% of patients achieved clinical improvement or cure at early post-therapy visit [40]. This is similar to our simulation results. Adding 2 g of SUL further increased maximal CFR to 73%. This is consistent with expert consensus on diagnosis and treatment of infection by Ab in China [41]: a daily dose of SUL should be increased to 6 g for treatment of infection by MDR, XDR, and PDR strains [42]. This is equivalent to the Q8h regimen of CPZ/SUL(2 g/2 g). ESCIM guidelines on the management and prevention of Ab infections also recommended that the daily dose of SUL should reach 9–12 g in three or four doses [43], which is equivalent to a Q8h or Q6h regimen of CPZ/SUL (2 g/3 g), respectively.

The effect of age on the PK/PD cutoff and CFR of sulbactam is shown in Appendix A and Appendix A. The impact of age was obvious. For an infusion of 2 h, when the age was 18, the PK/PD cutoff was 2 and 0.25 mg/L for clinical and microbiological efficacy, respectively. Corresponding CFR was 35.2% and 12.7%, respectively. When age increased to 70, the PK/PD cutoff elevated to 8 and 2 mg/L for clinical and microbiological efficacy, respectively. Corresponding CFR was 63.9% and 43.6%, respectively. This indicated that the effect of CPZ/SUL for the treatment of infection caused by Ab was better in the elderly compared to that in young patients. Meanwhile, compared to long infusion time, the impact of age was more obvious when infusion time was short. When infusion was 0.5 h, the PK/PD cutoff increased by seven times (4 vs. 0.5 mg/L) when age increased from 18 to 70 years (Appendix A). When infusion was 4 h, the PK/PD cutoff increased by one time (8 vs. 4 mg/L) when age increased from 18 to 70 years.

The shortcomings of this study were (1) PK interaction between CPZ and SUL could not be determined because all patients received the same dose ratio of CPZ/SUL (2:1). It has been reported that intravenous administration of CPZ and SUL separately do not show PK differences compared with a combination in healthy volunteers [44]. Whether this combination has PK interaction in HAP patients needs further evaluation; (2) number of patients included in this study is limited. A clinical trial recruiting more HAP patients is needed to confirm the findings of this study.

## 4. Materials and Methods

### 4.1. Study Design

A prospective, open label study was performed in Huashan Hospital affiliated with Fudan University and Yonghe Branch Hospital, China. The study protocol was approved by the Huashan Institutional Review Board and was registered at the Chinese Clinical Trial Registry (ChiCTR-OPN-16008848). The ethics of the study was consistent with the Declaration of Helsinki. Patients were recruited from two sites simultaneously. Informed consent was obtained from each patient or next of kin before the initiation of the study. Patients were enrolled if they met all inclusion criteria (Appendix A). If they met one of the exclusion criteria, they were excluded.

### 4.2. Drug

Sterile powder of Sulperazon for injection (1.5 g/vial) containing 1 g of CPZ and 0.5 g of SUL was provided by Pfizer Pharmaceutical Co., Ltd. (Dalian, China). The drug was dissolved using saline before use.

### 4.3. Dosing Regimen and Sample Collection

The dosing regimen was selected according to the sensitivity of isolated bacteria. Patients received an intravenous drip infusion of 3 g CPZ/SUL every 6, 8, or 12 h for 7–14 days. Infusion time was 1.5 h to 2 h. As cefoperazone could induce the lack of vitamin K which increases bleeding risk and hypoprothrombinemia [45], all patients received vitamin K1 10 mg daily to keep safe during treatment.

Patients were randomly assigned to four groups to perform sample collection (Appendix A). At each time point, a 4 mL blood sample was collected in lithium heparinized tubes. They were immediately separated using centrifugation at 3000 rpm (4 °C) for 10 min. The supernatant was transferred and was divided equally. All samples were stored at −70 °C. Plasma concentration of CPZ and SUL was determined using the liquid chromatography-tandem mass spectrometry method [46].

### 4.4. Clinical Observation and Antimicrobial Susceptibility

Before enrollment, demographic and baseline data including medical history, physical examination, bacterial test, and clinical laboratory tests of eligible patients were recorded. During the treatment, vital signs and clinical symptoms were evaluated daily. Bacterial tests, chest X-ray examinations, and clinical laboratory tests were followed up.

Bacteria were collected from blood or sputum culture, and the minimum inhibitory concentration (MIC) was determined using the agar dilution method according to CLSI guidelines [47]. For each isolate, escalating CPZ/SUL concentration in a fixed ratio (2:1) was employed to obtain MIC. *Escherichia coli* ATCC 25922 and *Pseudomonas aeruginosa* ATCC 27853 were used as quality control strains. The standard substances of cefoperazone sodium (titer 90.6%) and sulbactam sodium (titer 90.2%) were provided by Pfizer Co. Ltd. and the culture medium was Mueller-Hinton agar (OXOID Co. Ltd., Basingstoke, England).

### 4.5. Efficacy and Safety Evaluation

Clinical and microbiological efficacies were evaluated on day 2 and day 7 after the completion of drug treatment. The efficacy includes success and failure, where the definitions were summarized in Appendix A. All eligible patients were included in the safety analysis. Safety was evaluated by physical examination, vital signs, and safety laboratory tests. An adverse event (AE) was defined as any untoward medical occurrence which appeared or worsened during the study. AE was considered drug-related if it was classified as certainty, probably or possibly related to CPZ/SUL.

### 4.6. Population Pharmacokinetics

PPK models were developed based on the FOCEI method using Intel Fortran Compiler (Ver11.1, Intel Co., Ltd., City of Santa Clara, CA, USA), NONMEM (Ver7.1, ICON Development Solutions, Baltimore, MD, USA), PDxPop (Ver4.0, ICON Development Solutions, Baltimore, MD, USA), R (Ver4.1.2) [48] and PsN (Ver5.3, Uppsala University, Uppsala, Sweden). In addition to dose and concentration data, demographics, vital signs, blood routine and biochemistry values, treatment information, basic disease, and concomitant medications were collected as covariates.

Concentration-time profiles of CPZ and SUL were inspected to identify candidate structure models. Inter-individual variability (IIV) associated with structural model parameters were explored, as well as covariance between IIVs. As patients had PK sampling on more than one occasion (Appendix A), inter-occasion variability (IOV) was evaluated [49]. Parameterization using the notional structural model is: $P_{i}=TVP\cdot \exp(\eta_{P_{i}}+\kappa_{P_{i}})$, where *P*_i_ means the value of the structural parameter (*P)* in an individual (*i)*, *TVP* is a typical value, and *η*_Pi_ is IIV being normally distributed with mean zero and variance ω^2^. *κ*_Pi_ is IOV being normally distributed with mean zero and variance *π*^2^. Residual error was specified as a mixed model: $Y_{ij}=\hat{Y_{ij}}\cdot \left(1+\varepsilon_{1}\right)+\varepsilon_{2}$, where *Y*_ij_ and $\hat{Y_{ij}}$ are the observations and model prediction of drug concentration in individual *i* at time *j*, respectively. *ε*_1_ and *ε*_2_ are the proportional and additive error term, respectively. They are normally distributed with mean zero and variance σ^2^.

For continuous covariates, relationships between covariate and parameter *P* were modeled using power function [$P_{i}=TVP\cdot {\left(Cov/Cov_{median}\right)}^{\gamma_{P_{i}}}$] or linear function [$P_{i}=TVP\cdot \left(1+\gamma_{Pi}\cdot \left(Cov-Cov_{median}\right)\right)$], where *Cov* is the value of covariate for individual *i*, *Cov*_median_ is the median of the covariate. *γ*_Pi_ means exponent or coefficient. For binary or categorical covariates, effects of covariate were tested using power model: $P_{i}=TVP\cdot \theta_{P_{i}}{}^{Cov}$, where *θ*_Pi_ reflects the effect of the covariate on *P*. A stepwise forward inclusion and backward elimination process was used for covariate searching [50]. The inclusion of a covariate term was considered statistically significant if the objective function value (OFV) difference was greater than the critical value based on the χ^2^ test with an α of 0.01. Elimination of a covariate was performed if the *p*-value for OFV increase was higher than 0.008 (CPZ) or 0.001 (SUL).

The visual predictive check (VPC) was used to validate the PPK model [51]. The model simulation was performed 500 times using final estimates of the parameter. The bootstrap method was employed to evaluate the accuracy of model fittings [52]. This process was repeated 300 times.

Effects of covariates on PK parameters were performed using final parameter estimates. Each subject was simulated 40 times. The dosing regimen of CPZ/SUL was 3 g (q8h), and infusion time was 1.5 h. Based on simulated time profiles, non-compartment analysis was performed to obtain peak (C_max_) and trough concentration (C_min_), area under concentration time-curve (AUC_0-t_, AUC_0-∞_), half-life (T_1/2_), mean residence time (MRT_0-∞_), total clearance (CL_t_) and apparent distribution volume (V_d_) [53].

### 4.7. Pharmacokinetic/Pharmacodynamic Analysis

#### 4.7.1. Analysis Based on PK/PD Index for Single Drug

The idea of this part was the effect of the compound is classified as a single drug. PK/PD index (%T > MIC, AUC_0–24_/MIC, and C_max_/MIC) for CPZ were calculated using final estimates of parameters, where %T > MIC was calculated using differential equations. The correlation between each PK/PD index and clinical or microbiological efficacy was analyzed. Logistic regression and cross tabulation were used to obtain PK/PD target using R [49]. For SUL, the above was also performed. Monte Carlos simulation: the dose of CPZ included 1 g, 1.5 g, 2 g, 3 g, 4 g, and 6 g; the dose of SUL included 1 g, 1.5 g, 2 g, and 3 g; infusion time included 0.5 h, 1 h, 2 h, 3 h, and 4 h; dosing frequency was q8h. For each regimen, MCS was performed in 5400 patients using NONMEM. PTA was calculated as the probability of PK/PD index higher than the target. PK/PD cutoff was defined as maximal MIC with PTA ≥ 90% [54]. PK/PD cutoff for CPZ (2 g, q8h) and that for SUL (1 g, q8h) were compared. CFR was calculated according to $\sum_{i=1}^{n}\left(PTA_{i}\times F_{i}\right)$, where *F*_i_ was the distribution frequency of MIC, and *PTA*_i_ indicated PTA of the PK/PD index at *i*th MIC level.

#### 4.7.2. Analysis Based on Combined PK/PD Index

The combined PK/PD index was developed as follows: A + B or A × B, where A and B indicated the PK/PD index of CPZ and SUL, respectively. The PK/PD index with the highest correlation obtained from Section 4.7.1 was used to construct a combined PK/PD index. The correlation between a combined PK/PD index and clinical or microbiological efficacy was analyzed. The target for the combined PK/PD index was searched using logistic regression and cross tabulation. Monte Carlo simulation: dosing regimen of CPZ/SUL included: 1 g/1 g, 1.5 g/1.5 g, 2 g/1 g, 2 g/2 g, 2 g/3 g, 3 g/1 g, 4 g/1 g, and 6 g/1 g. Infusion time ranged from 0.5 h to 4 h. Frequency included q8h, q6h, and q12h. CFR, PTA, and the PK/PD cutoff was calculated using the method described in detail in Section 4.7.1.

#### 4.7.3. PK/PD Analysis Based on Joint PTA

The equation for joint PTA was: PTA(CPZ) × PTA(SUL), where PTA means the probability of target attainment of the PK/PD index for a single drug. The PK/PD index with the highest correlation with efficacy (obtained from Section 4.7.1) was selected in the calculation of joint PTA. MCS: dosing regimens tested were the same as Section 4.7.2. Joint PTA and PK/PD cutoff were calculated using the method described in Section 4.7.1.

## 5. Conclusions

To our knowledge, this is the first report on the population PK/PD of CPZ/SUL in HAP patients. Both PK of CPZ and SUL were consistent with the two-compartment model. High blood pressure was a covariate on PK of CPZ, while PK of SUL was affected by age and baclofen. Effects of high blood pressure and baclofen on PK were mild, indicating no requirement to adjust dosing regimens. For infection by Ab, the effect of CPZ/SUL was better in the elderly compared to that in young patients. Combined PK/PD index %(T > MIC_cpz_ × T > MIC_sul_), rather than PK/PD index for a single drug, was more suitable for characterization of PK/PD of CPZ/SUL. The main material basis for CPZ/SUL(2 g/1 g) was sulbactam. Probability of clinical efficacy reached 94% when %(T > MIC_cpz_ × T > MIC_sul_) ≥ 36.6%. Q8h regimen of CPZ/SUL (2 g/1 g) for 7–14 days showed good clinical efficacy when MIC_CPZ_ ≤ 4 mg/L. When MIC_CPZ_ was 8, 16, or 32 mg/L, CPZ/SUL 2 g/1 g (q8h), 2 g/2 g (q8h), or 2 g/2 g (q6h) (infusion was all 3 h) was recommended, respectively. A clinical efficacy and safety study to confirm simulation results is warranted.

## Figures and Tables

**Figure 1 antibiotics-11-00703-f001:**
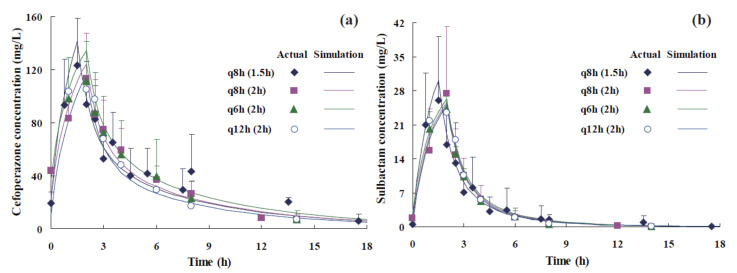
Time profiles of cefoperazone (**a**) and sulbactam (**b**) following the administration of 3 g of CPZ/SUL (2:1). Actual values are indicated using symbols (mean ± SD), while lines represent simulated values predicted by the PPK model. The brackets in legends are the infusion time.

**Figure 2 antibiotics-11-00703-f002:**
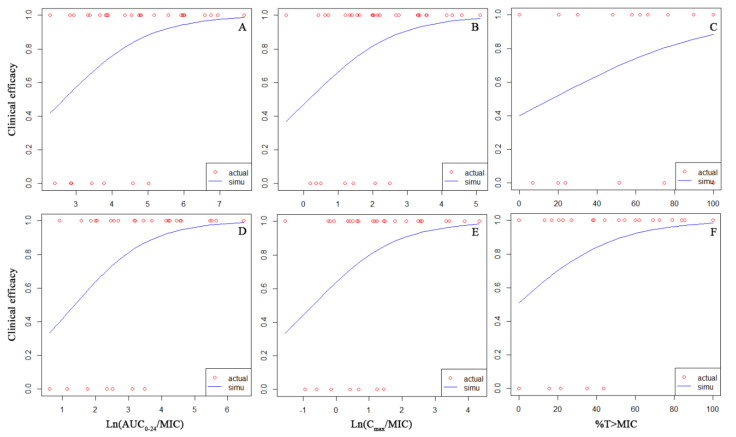
Correlation between the PK/PD index of cefoperazone (Panel **A**–**C**) or sulbactam (panel **D**–**F**) and clinical efficacy. Y-axis means the probability of successful clinical efficacy.

**Figure 3 antibiotics-11-00703-f003:**
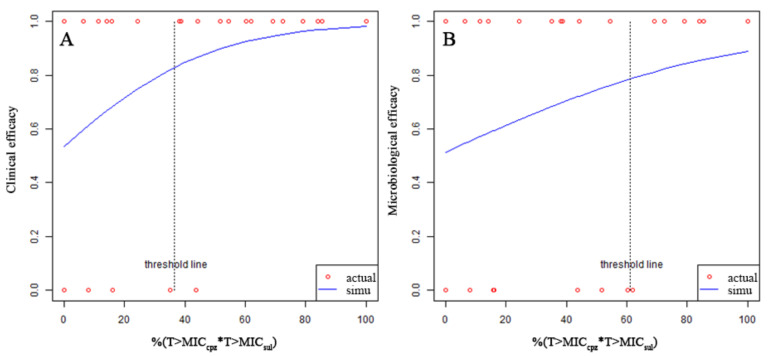
Correlation between the combined PK/PD index and clinical (**A**) or microbiological efficacy (**B**). The *p*-value of the slope obtained from logistic regression for clinical or microbiological efficacy were 0.047 and 0.114, respectively. cpz: cefoperazone; sul: sulbactam.

**Table 1 antibiotics-11-00703-t001:** Baseline characteristics of patients.

**Gender (Male/Female)**	39/15
**Age (yrs)**	46 ± 15 (18~70)
**Body Weight (kg)**	63.6 ± 8.3 (50~80)
**Body Mass Index (kg/m^2^)**	22.1 ± 2.1 (17.9~27.9)
**Concomitant Disease**	Traumatic brain injury: 24 cases (44%) Hypertension: 13 cases (24%) Cerebral hemorrhage: 11 cases (20%) Diabetes and other
**Alanine Aminotransferase**	Normal: 46 cases (85%) ≥1 time of upper normal limit (6 cases, 11%) ≥2 times of upper normal limit (2 cases, 4%)
**Renal Function**	147 ± 56 mL/min (52~342)
**Albumin**	Lower than normal: 16 cases (30%)
**Pathogen**	*Acinetobacter spp.*: 29 cases (54%) *P. aeruginosa*: 22 cases (41%) *Acinetobacter spp.* + *P. aeruginosa*: 3 cases (6%)
**Oral Temperature ≥ 37.5 °C**	49 cases (91%)
**White Blood Cells Count**	Abnormal: 28 cases (52%) Normal: 26 cases (48%)
**Oxygen Saturation**	52 cases (96%)
**Incision of Trachea**	50 cases (93%)
**Ventilator Assisted Breathing**	7 cases (13%)
**Antimicrobial Drug Treatment** **Two Weeks before Enrollment**	38 cases (70%)
**Concomitant Medications**	Ambroxol: 45 cases (83%) Sodium valproate: 27 cases (50%) Piracetam: 17 cases (31%)
**Concomitant Medications** **(Antifungal or Antibiotics)**	Fluconazole: 8 cases (15%) Vancomycin: 4 cases (7%)

Renal function was estimated using the Cockcroft–Gault equation.

**Table 2 antibiotics-11-00703-t002:** PPK model of cefoperazone/sulbactam and the parameter estimates.

Cefoperazone			Sulbactam		
Parameter	Estimate	Bootstrap	Parameter	Estimate	Bootstrap
CL (L/h)	4.45 (4.02)	4.45 (4.00)	CL (L/h)	15.8 (5.99)	15.8 (6.22)
V_1_ (L)	7.97 (29.4)	8.34 (11.7)	V_1_ (L)	18.0 (7.20)	17.9 (6.43)
Q (L/h)	12.03 (49.8)	11.05 (18.4)	Q (L/h)	2.91 (37.5)	2.92 (31.4)
V_2_ (L)	9.03 (19.4)	8.76 (10.5)	V_2_ (L)	5.39 (11.7)	5.41 (11.7)
γ_High BP on V2_	1.44 (13.4)	1.44 (14.2)	γ_age on CL_	−0.0152 (20.4)	−0.0145 (23.0)
			θ_baclofen on Q_	2.54 (33.8)	2.46 (26.5)
ω_CL_	0.226 (18.5)	0.224 (18.1)	ω_CL_	0.272 (23.2)	0.269 (24.3)
ω_V1_	0.448 (24.3)	0.441 (15.2)	ω_V1_	0.270 (18.1)	0.261 (18.2)
ω_V2_	0.337 (14.2)	0.319 (15.3)	ω_V2_	0.327 (22.7)	0.310 (26.8)
π_CL_	0.184 (21.2)	0.182 (21.0)	π_CL_	0.204 (21.6)	0.194 (25.1)
σ_Study in Huashan_	0.234 (15.6)	0.231 (18.0)	σ_Study in Huashan_	0.362 (17.4)	0.366 (18.8)
σ_Study in Yonghe_	0.102 (7.72)	0.102 (7.62)	σ_Study in Yonghe_	0.209 (11.5)	0.210 (10.5)
σ_add_	1.41 (36.1)	1.29 (26.0)	σ_add_	2.23 (38.6)	2.08 (68.8)
$\left\{\begin{array}{l} CL=4.45\;(L/h)\\ V_{1}=7.97\;(L)\\ Q=12.0\;(L/h)\\ V_{2}=9.03\times 1.44(if\;High\;BP)\;(L) \end{array}\right.$			$\left\{\begin{array}{l} CL=15.8\times [1-0.0152\times (Age-46.5)]\;(L/h)\\ V_{1}=18.0\;(L)\\ Q=2.91\times 2.54(if\;taking\;baclofen)\;(L/h)\\ V_{2}=5.39\;(L) \end{array}\right.$		

Parenthesis in the Estimate and Bootstrap columns indicate relative standard error (%). BP: blood pressure.

**Table 3 antibiotics-11-00703-t003:** PK/PD target cefoperazone and sulbactam against infection by Ab.

PK/PD Index	Cefoperazone		Sulbactam	
	Clinical Efficacy	Microbiological Efficacy	Clinical Efficacy	Microbiological Efficacy
AUC_0–24_/MIC	44.3 (90%)	162.4 (91%)	23.3 (94%)	50.4 (91%)
C_max_/MIC	3.29 (88%)	13.2 (91%)	3.46 (93%)	5.10 (91%)
%T > MIC	54.8% (88%)	83.2% (79%)	36.6% (94%)	61.1% (90%)

Value in parenthesis means probability of successful efficacy when PK/PD index ≥ target.

**Table 4 antibiotics-11-00703-t004:** PK/PD cutoff based on PK/PD index for a single drug (mg/L).

PK/PD Index	Drug	Regimen	Target	Infusion Time (h)				
				0.5	1	2	3	4
%T > MIC	cefoperazone	2 g q8h	54.8%	16	16	16	32	32
	sulbactam	1 g q8h	36.6%	1	2	2	4	4
AUC_0–24_/MIC	cefoperazone	2 g q8h	44.3	16	16	16	16	16
	sulbactam	1 g q8h	23.3	4	4	4	4	4

**Table 5 antibiotics-11-00703-t005:** Target for the combined PK/PD index against infection by Ab.

Combined PK/PD Index	Clinical Efficacy	Microbiological Efficacy
% (T > MIC_cpz_*T > MIC_sul_)	36.6% (94%)	61.1% (90%)
Ln(AUC_0–24_/MIC)_cpz_*Ln(AUC_0–24_/MIC)_sul_	14.37 (94%)	19.75 (91%)
%T > MIC_cpz_*Ln(AUC_0–24_/MIC)_sul_	3.15 (94%)	3.92 (91%)

Value in parenthesis means probability of successful efficacy when the PK/PD index ≥ target. cpz: cefoperazone; sul: sulbactam.

**Table 6 antibiotics-11-00703-t006:** PK/PD cutoff based on %(T > MIC_cpz_*T > MIC_sul_) against Ab (mg/L). Dosing frequency was q8h.

Regimen of Cefoperazone/Sulbactam	Clinical Efficacy					Microbiological Efficacy				
	T = 0.5	T = 1	T = 2	T = 3	T = 4	T = 0.5	T = 1	T = 2	T = 3	T = 4
1 g/1 g	2	2	4	8	8	0.5	0.5	1	2	4
2~6 g/1 g	2	4	4	8	8	0.5	0.5	1	2	4
1.5 g/1.5 g	4	4	8	8	16	1	1	2	2	4
2 g/2 g	4	4	8	16	16	1	1	2	4	8
2 g/3 g	8	8	16	16	16	2	2	4	4	8

Cutoff means maximal MIC of cefoperazone with the probability of target attainment ≥ 90%. T means infusion time (h).

**Table 7 antibiotics-11-00703-t007:** Cutoff based on PTA(%T > MIC_cpz_)*PTA(%T > MIC_sul_) (mg/L). Dosing frequency was q8h.

Regimen of Cefoperazone/Sulbactam	Clinical Efficacy					Microbiological Efficacy				
	T = 0.5	T = 1	T = 2	T = 3	T = 4	T = 0.5	T = 1	T = 2	T = 3	T = 4
2 g/1 g	2	4	4	8	8	0.5	0.5	1	2	4
2 g/2 g	4	8	8	16	16	1	1	2	4	8
2 g/3 g	8	8	16	16	32	2	2	2	4	8
4~6 g/1 g	2	4	4	8	8	0.5	0.5	1	2	4

The cutoff was for cefoperazone. T means infusion time (h). PTA: the probability of target attainment.

**Table 8 antibiotics-11-00703-t008:** Comparison of PK/PD cutoff obtained from combined PTA or single drug.

PK/PD Index	Type of Efficacy	Dosing Regimen of CPZ/SUL (q8h)				
		2 g/1 g	2 g/2 g	2 g/3 g	4 g/1 g	6 g/1 g
%T > MIC	Clinical efficacy	=SUL (2–8 mg/L)	=SUL (4–16 mg/L)	Close to SUL (8–32 mg/L)	=SUL (2–8 mg/L)	=SUL (2–8 mg/L)
	Microbiological efficacy	=SUL (0.5–4 mg/L)	=SUL (1–8 mg/L)	Close to SUL (2–8 mg/L)	=SUL (0.5–4 mg/L)	=SUL (0.5–4 mg/L)
AUC_0–24_/MIC	Clinical efficacy	=SUL (8 mg/L)	=SUL or CPZ (16 mg/L)	=SUL or CPZ (16 mg/L)	=SUL (8 mg/L)	=SUL (8 mg/L)
	Microbiological efficacy	=SUL or CPZ (4 mg/L)	=CPZ (4 mg/L)	=CPZ (4 mg/L)	=SUL (4 mg/L)	=SUL (4 mg/L)

=SUL: PK/PD cutoff based on the combined PTA = PK/PD cutoff based on sulbactam. Close to SUL: PK/PD cutoff based on combined PTA was close to PK/PD cutoff based on sulbactam. =SUL or CPZ: PK/PD cutoff based on combined PTA = PK/PD cutoff based on sulbactam or cefoperazone. =CPZ: PK/PD cutoff based on combined PTA = PK/PD cutoff based on cefoperazone. In parathesis, the range of PK/PD cutoff was obtained from combined PTA. The cutoff was for cefoperazone. Infusion time was 0.5–4 h.

## Data Availability

Data that support the findings of this study are available from the authors upon reasonable request.

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
