# Peer review of "Combined PK/PD Index May Be a More Appropriate PK/PD Index for Cefoperazone/Sulbactam against Acinetobacter baumannii in Patients with Hospital-Acquired Pneumonia"

_antibiotics, 2022, doi:10.3390/antibiotics11050703_

Round 1
Reviewer 1 Report
The authors of this paper performed PK/PD analysis on the cefioerazone/sulbactam combination in patients suffering from hospital-acquired pneumonia. While PK/PD studies on this beta lactam/inhibitor combo have been conducted for other conditions, data for pneumonia treatment are lacking. Overall this is a well conducted study and the results were presented well. One of the major concerns I have is regarding the patient population. Is there is a reason why majority of the patients in the study either had a brain injury or a cerebral hemorrhage? Do the authors envision any bias these underlying conditions place on the study itself?
Author Response
The authors of this paper performed PK/PD analysis on the cefoperazone/sulbactam combination in patients suffering from hospital-acquired pneumonia. While PK/PD studies on this beta lactam/inhibitor combo have been conducted for other conditions, data for pneumonia treatment are lacking. Overall this is a well conducted study and the results were presented well. One of the major concerns I have is regarding the patient population. Is there is a reason why majority of the patients in the study either had a brain injury or a cerebral hemorrhage?
Response: Thanks for your comment. Our response are as follows:
- Nosocomial infection is a common complication for cerebral hemorrhage. 36% of patients in neural ICU (stay time > 48h) have the complication of infection, where the pneumoniae is the most common [1]. Prof. Archana Hinduja et al performed a study to investigate the incidence rate of nosocomial infection in patients with intracerebral hemorrhage, as well as risk factor and prognosis. Results showed that 26% of patient have at least one infection, where the most common was pneumoniae (18%) [2].
- Severe craniocerebral trauma has been regarded as the risk factor for hospital-acquired and ventilator-acquired pneumoniae [3].
This has been added in the discussion. Please see line 288-296 (page 9).
Do the authors envision any bias these underlying conditions place on the study itself?
Response: Thanks for your comment. During population pharmacokinetic study, we have evaluated the impact of traumatic brain injury (variable name: DISCERTRA) and cerebral hemorrhage (variable name: DISCERHEM) on PK. Results showed that brain injury and cerebral hemorrhage did not have impact on PK of cefoperazone or sulbactam.
References
[1] Yang H, Fan Y, Li C, et al. A retrospective study on risk factors and disease burden for hospital-acquired pneumonia caused by multi-drug-resistant bacteria in patients with intracranial cerebral hemorrhage. Neurol Sci, 2022, 43(4): 2461-2467.
[2] Hinduja A, Dibu J, Achi E, et al. Nosocomial infections in patients with spontaneous intracerebral hemorrhage. Am J Crit Care, 2015, 24(3): 227-231.
[3] Infectious group of Respiratory branch of Chinese Medical Association. Guideline on diagnosis and treatment of hospital-acquired and ventilator-acquired pneumoniae in adults in China (2018 version). Chin J Tuberc Respir Dis, 2018, 41(4): 255-280.
Reviewer 2 Report
Multi-drug resistant and pan-drug resistant bacteria are becoming more and more prevalent due to the wide and non-judicious use of antibiotics. Treating different infections becomes more demanding each day passing. This multi-center epidemiological survey proved non-fermentative bacilli as the most common pathogen for hospital-acquired pneumonia (HAP) in China. Acinetobacter baumanii (Ab) and Pseudomonas aeruginosa (PA) constitute the majority of the isolated bacteria. As Cefoperazone/sulbactam combination of a β-lactam and β-lactamase inhibitor has been priorly selected as alternative drugs for the treatment of nonfermentative bacilli infection, this study aims to (1) describe PPK and pharmacodynamic (PD) of 73 CPZ/SUL in HAP patients;
(2) find the best PK/PD index predicting clinical and microbiological efficacy of CPZ/SUL;
(3) optimize the dosing regimen using Monte Carlo simulation.
The article contains valuable material but requires a few changes.
Point 1 – Please explain in the Introduction section the meaning of each acronym used.
Point 2 – define PD within the article when first mentioned, which might stand for pharmacodynamic.
Point 3 Fluconazole is an antifungal, not an antibiotic – please correct the terms on line 82. The same error is seen in Table 1 (Concomitant medications – antibiotics).
Point 4 The verb tense for the first proposition in Subsection 2.2 should be present time.
Point 5 Please define AE (line 106), and also what laboratory AE stands for?
Point 6 Please explain the meaning of IIV – line 109.
Point 7 As data from both study centers are cited in the Results section – Table 2, perhaps you should describe recruiting patients from two sites before describing this information. The reader should clearly understand at this point that patients were enrolled from Huashan Hospital, affiliated with Fudan University and Yonghe Branch Hospital, China.
Point 8 Spelling errors are spread throughout the text – see line 268 “becuase.”
Point 9 Please precisely tell if FOCE stands for The first-order conditional estimation method, firstly introduced in the text on line 298.
Point 10 Please precisely tell if FOCEI stands for the most commonly used estimation - First-order conditional estimation with interaction – line 300.
Point 11 Please acknowledge another limitation of this study – the limited number of patients included in the study group.
Point 12 Some of the cited references are quite old. I suggest consulting and citing the following articles:
https://doi.org/10.1016/j.jiph.2021.10.020
https://doi.org/10.3390/microorganisms9112384
https://doi.org/10.3390/antibiotics10091134
https://doi.org/10.3390/antibiotics9090597
Author Response
Point 1 – Please explain in the Introduction section the meaning of each acronym used.
Response: Thank you, we completed.
Point 2 – define PD within the article when first mentioned, which might stand for pharmacodynamic.
Response: Thank you, we completed. In line 65, "PK/PD" was changed into "pharmacokinetic/pharmacodynamic (PK/PD)".
Point 3 Fluconazole is an antifungal, not an antibiotic – please correct the terms on line 82. The same error is seen in Table 1 (Concomitant medications – antibiotics).
Response: Thanks for your comment. We have corrected them.
Point 4 The verb tense for the first proposition in Subsection 2.2 should be present time.
Response: Thank you, we corrected it. At line 94, 'were' was changed into 'are'.
Point 5 Please define AE (line 106), and also what laboratory AE stands for?
Response: Thanks for your comment. AE means adverse event. Lab AE means the abnormal level of laboratory examination indices (e.g., alanine aminotransferase, serum creatinine, glucose). At line 107-108, the sentence was changed into ‘No laboratory adverse event (AE) or serious AE was observed’.
Point 6 Please explain the meaning of IIV – line 109.
Response: Thanks for your comment. IIV means inter-individual variability. We add the meaning of IIV at line 110.
Point 7 As data from both study centers are cited in the Results section – Table 2, perhaps you should describe recruiting patients from two sites before describing this information. The reader should clearly understand at this point that patients were enrolled from Huashan Hospital, affiliated with Fudan University and Yonghe Branch Hospital, China.
Response: Thanks for your suggestion. We add a sentence (Patients were recruited from both two sites simultaneously) in section 4.1 (line 479, page 13).
Point 8 Spelling errors are spread throughout the text – see line 268 “becuase.”
Response: Thanks for your comment. We correct it. Please see line 267 (page 9).
Point 9 Please precisely tell if FOCE stands for the first-order conditional estimation method, firstly introduced in the text on line 298.
Response: Thanks for your comment. Yes, FOCE stands for first-order conditional estimation. We add the meaning of FOCE on line 306 (page 10).
Point 10 Please precisely tell if FOCEI stands for the most commonly used estimation - First-order conditional estimation with interaction – line 300.
Response: Thanks for your comment. We add the meaning of FOCI (first-order conditional estimation with interaction) on line 308-309 (page 10).
Point 11 Please acknowledge another limitation of this study – the limited number of patients included in the study group.
Response: Thanks for your comments. We add two sentences (Number of patients included in this study is limited. A clinical trial recruiting more HAP patients is needed to confirm the findings of this study) on line 469-471 (page 13).
Point 12 Some of the cited references are quite old. I suggest consulting and citing the following articles:
https://doi.org/10.1016/j.jiph.2021.10.020 (Antibiotic resistance in microbes: History, mechanisms, therapeutic strategies and future prospects)
https://doi.org/10.3390/microorganisms9112384 (Relationship between the Biofilm-Forming Capacity and Antimicrobial Resistance in Clinical Acinetobacter baumannii Isolates: Results from a Laboratory-Based In Vitro Study)
https://doi.org/10.3390/antibiotics10091134 (No Correlation between Biofilm Formation, Virulence Factors, and Antibiotic Resistance in Pseudomonas aeruginosa: Results from a Laboratory-Based In Vitro Study)
https://doi.org/10.3390/antibiotics9090597 (Evidence of the Practice of Self-Medication with Antibiotics among the Lay Public in Low- and Middle-Income Countries: A Scoping Review)
Response: Thank you for your suggestion. We downloaded these articles. The first article (Antibiotic resistance in microbes: History, mechanisms, therapeutic strategies and future prospects, https://doi.org/10.1016/j.jiph.2021.10.020) and the last article (Evidence of the Practice of Self-Medication with Antibiotics among the Lay Public in Low- and Middle-Income Countries: A Scoping Review, https://doi.org/10.3390/antibiotics9090597) have been cited on line 42 (Introduction, first paragraph). Please see reference 2-3 (page16-17, line 646-651).
Reviewer 3 Report
In this manuscript, Zhou et al developed a PPK and PD of CPZ/SUL in HAP patients, identified the best PK/PD index for CPZ/SUL, and optimized the dosing regimen using Monte Carlo simulation. The manuscript is well written and is clinically meaningful. The authors identified that the combined PK/PD index is more suitable than the PK/PD index for a single drug and the appropriate dosing regimes were also proposed for each MIC of CPZ. This study has several strengths including a prospective multicenter design, a well-structured methodology (inclusion/exclusion criteria, PPK model development, and evaluation). Some comments are listed below and should be addressed before being published.
- The authors provided a background for the PK of CPZ/SUL. Are there any reports on the PPK-PD of this compound?
- While the methodology and results on model development and evaluation were well described, the discussion on significant covariates was limited. The authors should compare the results (significant covariates) with other studies. Also, supporting explanations/reasons on how the covariates influenced PK parameters should also be provided.
- Are there any explanations for why different estimation methods resulted in different significant covariates?
- Was the cutoff for CPZ/SUL different for different patient populations?
Author Response
- The authors provided a background for the PK of CPZ/SUL. Are there any reports on the PPK-PD of this compound?
Response: Thanks for your question. We searched the literature. There are some PK/PD report of CPZ/SUL [1-4]. Meanwhile, there are two reports on the PPK-PD of CPZ in children [5; 6]. However, up to now, there are no reports on PPK-PD of CPZ/SUL. We add a sentence in the introduction. Please see line 62-63 (page 2).
- While the methodology and results on model development and evaluation were well described, the discussion on significant covariates was limited.
Response: Thanks for your comment. We add two paragraphs to discuss significant covariates on the PK of CPZ/SUL (page 10, line 320-331). The content was as follows:
Results showed that high blood pressure have significant impact on V2 of cefoperazone. The explanation was as follows: (1) Cefoperazone is a lipophilic drug. It could distribute to peripheral compartment (i.e., peripheral tissue) easily; (2) Cerebral hemorrhage is one of the main complications of high blood pressure [7]. One study also showed that high blood pressure is strong risk factor for hemorrhagic stroke [8]. In this study, many patients had cerebral hemorrhage (20%) or brain injury (44%) (Table 2). So, these diseases may destroy the barrier between blood and peripheral tissue. Hence, cefoperazone could easily reach peripheral tissue in the patients with high blood pressure.
Sulbactam is a water-soluble drug. After administration, approximately 75%-84% of the dose was excreted by kidney [9; 10]. When the age increases, the renal function decreases. Therefore, the clearance of sulbactam also decreases. This could explain why age has significant impact on clearance of sulbactam.
Impact of baclofen on inter-compartment clearance (Q) of sulbactam had been discussed in the manuscript on line 337-340 (page 10).
- The authors should compare the results (significant covariates) with other studies.
Response: Thanks for your comment. In the discussion, we add the comparison of PK results (significant covariates) of sulbactam with other studies (page 10, line 330-336). The content was as follows:
Our study showed that age has significant impact on CL of sulbactam. This was consistent with literature reports: A study in Thailand showed that age and CLcr are covariates on CL of sulbactam, while HgB was covariate on V1 [11]. Another work showed that age had significant impact on PK of sulbactam [12]: compared to younger (20-40 years) or middle-aged subjects (41-64 years), the AUC (P<0.05), Cmax and T1/2 (P<0.05) of sulbactam was the greatest in elderly subjects (65-85 years), while the CL was the lowest (P<0.05).
For the cefoperazone, there are two PPK reports in children [5; 6]. There are no PPK report in adults. Because many aspects are different between children and adults, we did not compare the PPK results of cefoperazone between adults and children.
- Also, supporting explanations/reasons on how the covariates influenced PK parameters should also be provided.
Response: Thanks for your comment. We add two paragraphs in the discussion (page 10, line 320-336). They provide explanation on how high blood pressure influence V2 of CPZ, and on how the age influence the CL of SUL. The reason on how the baclofen influence the Q of SUL had been provided on line 337-340 (page 10).
- Are there any explanations for why different estimation methods resulted in different significant covariates?
Response: Thanks for your comment. FOCE does not consider the interaction between intra- and inter-individual variability. However, FOCEI considers this interaction and performs calculation [13; 14]. So, the FOCEI method could improve the model fit because it could explain part of variability in the data. Hence, for the same covariate, compared to the FOCE method, reduction of OBJ was different when using FOCEI method. This could result different significant covariates in the PPK study. This explanation has been added in the discussion (page 10, line 314-319).
- Was the cutoff for CPZ/SUL different for different patient populations?
Response: Thanks for your question. Our results showed that age have impact on the cutoff of CPZ/SUL. The cutoff elevated along with increased age (Table S15). For the regimen CPZ/SUL(2g/1g) (q8h), when infusion 2h, the cutoff of CPZ/SUL was 4/2 and 16/8 mg/L in the HAP patients with age 18 and 70, respectively.
References
- Ye, L.; Cheng, L.; Kong, L.; Zhao, X.; Xie, G.; He, J.; Liu, H.; Deng, Y.; Wu, X.; Wang, T.; Yang, X. Pharmacokinetic and pharmacodynamic analysis of cefoperazone/sulbactam for the treatment of pediatric sepsis by Monte Carlo simulation. Anal Methods 2022, 14, 1148-1154.
- Dong, Y.; Li, Y.; Zhang, Y.; Zhang, T.; Zhu, L.; Dong, Y.; Wang, T. Cefoperazone/sulbactam therapeutic drug monitoring in patients with liver cirrhosis: Potential factors affecting the pharmacokinetic/pharmacodynamic target attainment. Basic Clin Pharmacol Toxicol 2019, 125, 353-359.
- Zhu, W.; Chu, Y.; Zhang, J.; Xian, W.; Xu, X.; Liu, H. Pharmacokinetic and pharmacodynamic profiling of four antimicrobials against Acinetobacter baumannii infection. Microb. Pathog. 2020, 138, 103809.
- Xiao, Y. H.; Hu, Y. J. The reliability of using impenem, meropenem, cefoperazone-sulbactam and piperacillin-tazobactam to treat nosocomial Gram-negative bacterial infections with Monte Carlo simulation. Zhonghua nei ke za zhi 2017, 56, 595-600.
- Xing, L. P.; Niu, C. H.; Huang, R.; Gao, L. L.; Yu, L. P.; Wang, Y.; Li, J. The population pharmacokinetics and dose optimization of cefoperazone sodium and sulbactam sodium in children. Chin J Hosp Pharm 2020, 40, 2102-2007,2142.
- Shi, H. Y.; Wang, K.; Wang, R. H.; Wu, Y. E.; Tang, B. H.; Li, X.; Du, B.; Kan, M.; Zheng, Y.; Xu, B. P.; Shen, A. D.; Su, L. Q.; Jacqz-Aigrain, E.; Huang, X.; Zhao, W. Developmental population pharmacokinetics-pharmacodynamics and dosing optimization of cefoperazone in children. J Antimicrob Chemother 2020, 75, 1917-1924.
- Lattanzi, S.; Silvestrini, M. Blood pressure in acute intra-cerebral hemorrhage. Ann Transl Med 2016, 4, 320.
- Hagg-Holmberg, S.; Dahlstrom, E. H.; Forsblom, C. M.; Harjutsalo, V.; Liebkind, R.; Putaala, J.; Tatlisumak, T.; Groop, P. H.; Thorn, L. M.; FinnDiane Study, G. The role of blood pressure in risk of ischemic and hemorrhagic stroke in type 1 diabetes. Cardiovasc. Diabetol. 2019, 18, 88.
- Foulds, G.; Stankewich, J. P.; Marshall, D. C.; O'Brien, M. M.; Hayes, S. L.; Weidler, D. J.; McMahon, F. G. Pharmacokinetics of sulbactam in humans. Antimicrob Agents Chemother 1983, 23, 692-699.
- Pfizer, Package insert for Cefoperazone Sodium and Sulbactam Sodium for Injection (20220329 Version), Dalian, 2022. Available online: https://labeling.pfizer.com/ShowLabeling.aspx?id=14442. (accessed on 2022-05-18).
- Jaruratanasirikul, S.; Wongpoowarak, W.; Wattanavijitkul, T.; Sukarnjanaset, W.; Samaeng, M.; Nawakitrangsan, M.; Ingviya, N. Population Pharmacokinetics and Pharmacodynamics Modeling To Optimize Dosage Regimens of Sulbactam in Critically Ill Patients with Severe Sepsis Caused by Acinetobacter baumannii. Antimicrob Agents Chemother 2016, 60, 7236-7244.
- Meyers, B. R.; Wilkinson, P.; Mendelson, M. H.; Walsh, S.; Bournazos, C.; Hirschman, S. Z. Pharmacokinetics of ampicillin-sulbactam in healthy elderly and young volunteers. Antimicrob Agents Chemother 1991, 35, 2098-2101.
- Wang, Y. Derivation of various NONMEM estimation methods. J Pharmacokinet Pharmacodyn 2007, 34, 575-593.
- Bauer, R. J. NONMEM Tutorial Part II: Estimation Methods and Advanced Examples. CPT Pharmacometrics Syst Pharmacol 2019, 8, 538-556.